# Multi-Modal Imitation Learning from Unstructured Demonstrations using Generative Adversarial Nets

**Karol Hausman**[*†], **Yevgen Chebotar**[*†‡], **Stefan Schaal**[†‡], **Gaurav Sukhatme**[†], **Joseph J. Lim**[†]

[†]University of Southern California, Los Angeles, CA, USA
[‡]Max-Planck-Institute for Intelligent Systems, Tübingen, Germany
{hausman, ychebota, sschaal, gaurav, limjj}@usc.edu

## Abstract

Imitation learning has traditionally been applied to learn a single task from demonstrations thereof. The requirement of *structured* and isolated demonstrations limits the scalability of imitation learning approaches as they are difficult to apply to real-world scenarios, where robots have to be able to execute a multitude of tasks. In this paper, we propose a multi-modal imitation learning framework that is able to segment and imitate skills from unlabelled and *unstructured* demonstrations by learning skill segmentation and imitation learning jointly. The extensive simulation results indicate that our method can efficiently separate the demonstrations into individual skills and learn to imitate them using a single multi-modal policy. The video of our experiments is available at http://sites.google.com/view/nips17intentiongan.

## 1 Introduction

One of the key factors to enable deployment of robots in unstructured real-world environments is their ability to learn from data. In recent years, there have been multiple examples of robot learning frameworks that present promising results. These include: reinforcement learning [31] - where a robot learns a skill based on its interaction with the environment and imitation learning [2, 5] - where a robot is presented with a demonstration of a skill that it should imitate. In this work, we focus on the latter learning setup.

Traditionally, imitation learning has focused on using isolated demonstrations of a particular skill [29]. The demonstration is usually provided in the form of kinesthetic teaching, which requires the user to spend sufficient time to provide the right training data. This constrained setup for imitation learning is difficult to scale to real world scenarios, where robots have to be able to execute a combination of different skills. To learn these skills, the robots would require a large number of robot-tailored demonstrations, since at least one isolated demonstration has to be provided for every individual skill.

In order to improve the scalability of imitation learning, we propose a framework that can learn to imitate skills from a set of unstructured and unlabeled demonstrations of various tasks.

As a motivating example, consider a highly unstructured data source, e.g. a video of a person cooking a meal. A complex activity, such as cooking, involves a set of simpler skills such as grasping, reaching, cutting, pouring, etc. In order to learn from such data, three components are required: i) the ability to map the image stream to state-action pairs that can be executed by a robot, ii) the ability to segment the data into simple skills, and iii) the ability to imitate each of the segmented skills. In this work, we tackle the latter two components, leaving the first one for future work. We believe that the capability proposed here of learning from unstructured, unlabeled demonstrations is an important step towards scalable robot learning systems.

---

[*]Equal contribution

In this paper, we present a novel imitation learning method that learns a multi-modal stochastic policy, which is able to imitate a number of automatically segmented tasks using a set of unstructured and unlabeled demonstrations. Our results indicate that the presented technique can separate the demonstrations into sensible individual skills and imitate these skills using a learned multi-modal policy. We show applications of the presented method to the tasks of skill segmentation, hierarchical reinforcement learning and multi-modal policy learning.

## 2  Related Work

Imitation learning is concerned with learning skills from demonstrations. Approaches that are suitable for this setting can be split into two categories: i) behavioral cloning [27], and ii) inverse reinforcement learning (IRL) [24]. While behavioral cloning aims at replicating the demonstrations exactly, it suffers from the covariance shift [28]. IRL alleviates this problem by learning a reward function that explains the behavior shown in the demonstrations. The majority of IRL works [16, 35, 1, 12, 20] introduce algorithms that can imitate a single skill from demonstrations thereof but they do not readily generalize to learning a multi-task policy from a set of unstructured demonstrations of various tasks.

More recently, there has been work that tackles a problem similar to the one presented in this paper, where the authors consider a setting where there is a large set of tasks with many instantiations [10]. In their work, the authors assume a way of communicating a new task through a single demonstration. We follow the idea of segmenting and learning different skills jointly so that learning of one skill can accelerate learning to imitate the next skill. In our case, however, the goal is to separate the mix of expert demonstrations into single skills and learn a policy that can imitate all of them, which eliminates the need of new demonstrations at test time.

The method presented here belongs to the field of multi-task inverse reinforcement learning. Examples from this field include [9] and [4]. In [9], the authors present a Bayesian approach to the problem, while the method in [4] is based on an EM approach that clusters observed demonstrations. Both of these methods show promising results on relatively low-dimensional problems, whereas our approach scales well to higher dimensional domains due to the representational power of neural networks.

There has also been a separate line of work on learning from demonstration, which is then iteratively improved through reinforcement learning [17, 6, 23]. In contrast, we do not assume access to the expert reward function, which is required to perform reinforcement learning in the later stages of the above algorithms.

There has been much work on the problem of skill segmentation and option discovery for hierarchical tasks. Examples include [25, 19, 14, 33, 13]. In this work, we consider a possibility to discover different skills that can all start from the same initial state, as opposed to hierarchical reinforcement learning where the goal is to segment a task into a set of consecutive subtasks. We demonstrate, however, that our method may be used to discover the hierarchical structure of a task similarly to the hierarchical reinforcement learning approaches. In [13], the authors explore similar ideas to discover useful skills. In this work, we apply some of these ideas to the imitation learning setup as opposed to the reinforcement learning scenario.

Generative Adversarial Networks (GANs) [15] have enjoyed success in various domains including image generation [8], image-image translation [34, 18] and video prediction [22]. More recently, there have been works connecting GANs and other reinforcement learning and IRL methods [26, 11, 16]. In this work, we expand on some of the ideas presented in these works and provide a novel framework that exploits this connection.

The works that are most closely related to this paper are [16], [7] and [21]. In [7], the authors show a method that is able to learn disentangled representations and apply it to the problem of image generation. In this work, we provide an alternative derivation of our method that extends their work and applies it to multi-modal policies. In [16], the authors present an imitation learning GAN approach that serves as a basis for the development of our method. We provide an extensive evaluation of the hereby presented approach compared to the work in [16], which shows that our method, as opposed to [16], can handle unstructured demonstrations of different skills. A concurrent work [21] introduces a method similar to ours and applies it to detecting driving styles from unlabelled human data.

# 3 Preliminaries

Let $\mathcal{M} = (S, A, P, R, p_0, \gamma, T)$ be a finite-horizon Markov Decision Process (MDP), where $S$ and $A$ are state and action spaces, $P : S \times A \times S \to \mathbb{R}_+$ is a state-transition probability function or system dynamics, $R : S \times A \to \mathbb{R}$ a reward function, $p_0 : S \to \mathbb{R}_+$ an initial state distribution, $\gamma$ a reward discount factor, and $T$ a horizon. Let $\tau = (s_0, a_0, \ldots, s_T, a_T)$ be a trajectory of states and actions and $R(\tau) = \sum_{t=0}^{T} \gamma^t R(s_t, a_t)$ the trajectory reward. The goal of reinforcement learning methods is to find parameters $\theta$ of a policy $\pi_\theta(a|s)$ that maximizes the expected discounted reward over trajectories induced by the policy: $\mathbb{E}_{\pi_\theta}[R(\tau)]$ where $s_0 \sim p_0, s_{t+1} \sim P(s_{t+1}|s_t, a_t)$ and $a_t \sim \pi_\theta(a_t|s_t)$.

In an imitation learning scenario, the reward function is unknown. However, we are given a set of demonstrated trajectories, which presumably originate from some optimal expert policy distribution $\pi_{E_1}$ that optimizes an unknown reward function $R_{E_1}$. Thus, by trying to estimate the reward function $R_{E_1}$ and optimizing the policy $\pi_\theta$ with respect to it, we can recover the expert policy. This approach is known as inverse reinforcement learning (IRL) [1]. In order to model a variety of behaviors, it is beneficial to find a policy with the highest possible entropy that optimizes $R_{E_1}$. We will refer to this approach as the maximum-entropy IRL [35] with the optimization objective

$$\min_R \left( \max_{\pi_\theta} H(\pi_\theta) + \mathbb{E}_{\pi_\theta} R(s, a) \right) - \mathbb{E}_{\pi_{E_1}} R(s, a), \tag{1}$$

where $H(\pi_\theta)$ is the entropy of the policy $\pi_\theta$.

Ho and Ermon [16] showed that it is possible to redefine the maximum-entropy IRL problem with multiple demonstrations sampled from a single expert policy $\pi_{E_1}$ as an optimization of GANs [15]. In this framework, the policy $\pi_\theta(a|s)$ plays the role of a generator, whose goal is to make it difficult for a discriminator network $D_w(s, a)$ (parameterized by $w$) to differentiate between imitated samples from $\pi_\theta$ (labeled 0) and demonstrated samples from $\pi_{E_1}$ (labeled 1). Accordingly, the joint optimization goal can be defined as

$$\max_\theta \min_w \mathbb{E}_{(s,a) \sim \pi_\theta}[\log(D_w(s, a))] + \mathbb{E}_{(s,a) \sim \pi_{E_1}}[\log(1 - D_w(s, a))] + \lambda_H H(\pi_\theta). \tag{2}$$

The discriminator and the generator policy are both represented as neural networks and optimized by repeatedly performing alternating gradient updates. The discriminator is trained on the mixed set of expert and generator samples and outputs probabilities that a particular sample has originated from the generator or the expert policies. This serves as a reward signal for the generator policy that tries to maximize the probability of the discriminator confusing it with an expert policy. The generator can be trained using the trust region policy optimization (TRPO) algorithm [30] with the cost function $\log(D_w(s, a))$. At each iteration, TRPO takes the following gradient step:

$$\mathbb{E}_{(s,a) \sim \pi_\theta}[\nabla_\theta \log \pi_\theta(a|s) \log(D_w(s, a))] + \lambda_H \nabla_\theta H(\pi_\theta), \tag{3}$$

which corresponds to minimizing the objective in Eq. (2) with respect to the policy $\pi_\theta$.

# 4 Multi-modal Imitation Learning

The traditional imitation learning scenario described in Sec. 3 considers a problem of learning to imitate one skill from demonstrations. The demonstrations represent samples from a single expert policy $\pi_{E1}$. In this work, we focus on an imitation learning setup where we learn from unstructured and unlabelled demonstrations of various tasks. In this case, the demonstrations come from a set of expert policies $\pi_{E_1}, \pi_{E_2}, \ldots, \pi_{E_k}$, where $k$ can be unknown, that optimize different reward functions/tasks. We will refer to this set of unstructured expert policies as a mixture of policies $\pi_E$. We aim to segment the demonstrations of these policies into separate tasks and learn a multi-modal policy that will be able to imitate all of the segmented tasks.

In order to be able to learn multi-modal policy distributions, we augment the policy input with a latent intention $i$ distributed by a categorical or uniform distribution $p(i)$, similar to [7]. The goal of the intention variable is to select a specific mode of the policy, which corresponds to one of the skills presented in the demonstrations. The resulting policy can be expressed as:

$$\pi(a|s, i) = p(i|s, a) \frac{\pi(a|s)}{p(i)}. \tag{4}$$

We augment the trajectory to include the latent intention as $\tau_i = (s_0, a_0, i_0, ...s_T, a_T, i_T)$. The resulting reward of the trajectory with the latent intention is $R(\tau_i) = \sum_{t=0}^{T} \gamma^t R(s_t, a_t, i_t)$. $R(a, s, i)$ is a reward function that depends on the latent intention $i$ as we have multiple demonstrations that optimize different reward functions for different tasks. The expected discounted reward is equal to: $\mathbb{E}_{\pi_\theta}[R(\tau_i)] = \int R(\tau_i)\pi_\theta(\tau_i)d\tau_i$ where $\pi_\theta(\tau_i) = p_0(s_0)\prod_{t=0}^{T-1} P(s_{t+1}|s_t, a_t)\pi_\theta(a_t|s_t, i_t)p(i_t)$.

Here, we show an extension of the derivation presented in [16] (Eqs. (1, 2)) for a policy $\pi(a|s, i)$ augmented with the latent intention variable $i$, which uses demonstrations from a set of expert policies $\pi_E$, rather than a single expert policy $\pi_{E_1}$. We are aiming at maximum entropy policies that can be determined from the latent intention variable $i$. Accordingly, we transform the original IRL problem to reflect this goal:

$$\min_{R} \left( \max_{\pi} H(\pi(a|s)) - H(\pi(a|s, i)) + \mathbb{E}_\pi R(s, a, i) \right) - \mathbb{E}_{\pi_E} R(s, a, i), \tag{5}$$

where $\pi(a|s) = \sum_i \pi(a|s, i)p(i)$, which results in the policy averaged over intentions (since the $p(i)$ is constant). This goal reflects our objective: we aim to obtain a multi-modal policy that has a high entropy without any given intention, but it collapses to a particular task when the intention is specified. Analogously to the solution for a single expert policy, this optimization objective results in the optimization goal of the generative adversarial imitation learning network, with the exception that the state-action pairs $(s, a)$ are sampled from a set of expert policies $\pi_E$:

$$\max_{\theta} \min_{w} \mathbb{E}_{i \sim p(i), (s,a) \sim \pi_\theta}[\log(D_w(s, a))] + \mathbb{E}_{(s,a) \sim \pi_E}[1 - \log(D_w(s, a))] \tag{6}$$
$$+ \lambda_H H(\pi_\theta(a|s)) - \lambda_I H(\pi_\theta(a|s, i)),$$

where $\lambda_I$, $\lambda_H$ correspond to the weighting parameters on the respective objectives. The resulting entropy $H(\pi_\theta(a|s, i))$ term can be expressed as:

$$H(\pi_\theta(a|s, i)) = \mathbb{E}_{i \sim p(i), (s,a) \sim \pi_\theta}(-\log(\pi_\theta(a|s, i))) \tag{7}$$
$$= -\mathbb{E}_{i \sim p(i), (s,a) \sim \pi_\theta} \log \left( p(i|s, a)\frac{\pi_\theta(a|s)}{p(i)} \right)$$
$$= -\mathbb{E}_{i \sim p(i), (s,a) \sim \pi_\theta} \log(p(i|s, a)) - \mathbb{E}_{i \sim p(i), (s,a) \sim \pi_\theta} \log \pi_\theta(a|s) + \mathbb{E}_{i \sim p(i)} \log p(i)$$
$$= -\mathbb{E}_{i \sim p(i), (s,a) \sim \pi_\theta} \log(p(i|s, a)) + H(\pi_\theta(a|s)) - H(i),$$

which results in the final objective:

$$\max_{\theta} \min_{w} \mathbb{E}_{i \sim p(i), (s,a) \sim \pi_\theta}[\log(D_w(s, a))] + \mathbb{E}_{(s,a) \sim \pi_E}[1 - \log(D_w(s, a))] \tag{8}$$
$$+ (\lambda_H - \lambda_I)H(\pi_\theta(a|s)) + \lambda_I \mathbb{E}_{i \sim p(i), (s,a) \sim \pi_\theta} \log(p(i|s, a)) + \lambda_I H(i),$$

where $H(i)$ is a constant that does not influence the optimization. This results in the same optimization objective as for the single expert policy (see Eq. (2)) with an additional term $\lambda_I \mathbb{E}_{i \sim p(i), (s,a) \sim \pi_\theta} \log(p(i|s, a))$ responsible for rewarding state-action pairs that make the latent intention inference easier. We refer to this cost as the latent intention cost and represent $p(i|s, a)$ with a neural network. The final reward function for the generator is:

$$\mathbb{E}_{i \sim p(i), (s,a) \sim \pi_\theta}[\log(D_w(s, a))] + \lambda_I \mathbb{E}_{i \sim p(i), (s,a) \sim \pi_\theta} \log(p(i|s, a)) + \lambda_{H'} H(\pi_\theta(a|s)). \tag{9}$$

## 4.1 Relation to InfoGAN

In this section, we provide an alternative derivation of the optimization goal in Eq. (8) by extending the InfoGAN approach presented in [7]. Following [7], we introduce the latent variable $c$ as a means to capture the semantic features of the data distribution. In this case, however, the latent variables are used in the imitation learning scenario, rather than the traditional GAN setup, which prevents us from using additional noise variables ($z$ in the InfoGAN approach) that are used as noise samples to generate the data from.

Similarly to [7], to prevent collapsing to a single mode, the policy optimization objective is augmented with mutual information $I(c; G(\pi_\theta^c, c))$ between the latent variable and the state-action pairs generator $G$ dependent on the policy distribution $\pi_\theta^c$. This encourages the policy to produce behaviors that are

interpretable from the latent code, and given a larger number of possible latent code values leads to an increase in the diversity of policy behaviors. The corresponding generator goal can be expressed as:

$$\mathbb{E}_{c \sim p(c), (s,a) \sim \pi_\theta^c}[\log(D_w(s,a))] + \lambda_I I(c; G(\pi_\theta^c, c)) + \lambda_H H(\pi_\theta^c) \qquad (10)$$

In order to compute $I(c; G(\pi_\theta^c, c))$, we follow the derivation from [7] that introduces a lower bound:

$$
\begin{aligned}
I(c; G(\pi_\theta^c, c)) &= H(c) - H(c|G(\pi_\theta^c, c)) \qquad (11) \\
&= \mathbb{E}_{(s,a) \sim G(\pi_\theta^c, c)}[\mathbb{E}_{c' \sim P(c|s,a)}[\log P(c'|s,a)]] + H(c) \\
&= \mathbb{E}_{(s,a) \sim G(\pi_\theta^c, c)}[D_{KL}(P(\cdot|s,a)||Q(\cdot|s,a)) + \mathbb{E}_{c' \sim P(c|s,a)}[\log Q(c'|s,a)]] + H(c) \\
&\geq \mathbb{E}_{(s,a) \sim G(\pi_\theta^c, c)}[\mathbb{E}_{c' \sim P(c|s,a)}[\log Q(c'|s,a)]] + H(c) \\
&= \mathbb{E}_{c \sim P(c), (s,a) \sim G(\pi_\theta^c, c)}[\log Q(c|s,a)] + H(c)
\end{aligned}
$$

By maximizing this lower bound we maximize $I(c; G(\pi_\theta^c, c))$. The auxiliary distribution $Q(c|s,a)$ can be parametrized by a neural network.

The resulting optimization goal is

$$
\begin{aligned}
\max_\theta \min_w \; &\mathbb{E}_{c \sim p(c), (s,a) \sim \pi_\theta^c}[\log(D_w(s,a))] + \mathbb{E}_{(s,a) \sim \pi_E}[1 - \log(D_w(s,a))] \qquad (12) \\
&+ \lambda_I \mathbb{E}_{c \sim P(c), (s,a) \sim G(\pi_\theta^c, c)}[\log Q(c|s,a)] + \lambda_H H(\pi_\theta^c)
\end{aligned}
$$

which results in the generator reward function:

$$\mathbb{E}_{c \sim p(c), (s,a) \sim \pi_\theta^c}[\log(D_w(s,a))] + \lambda_I \mathbb{E}_{c \sim P(c), (s,a) \sim G(\pi_\theta^c, c)}[\log Q(c|s,a)] + \lambda_H H(\pi_\theta^c). \qquad (13)$$

This corresponds to the same objective that was derived in Section 4. The auxiliary distribution over the latent variables $Q(c|s,a)$ is analogous to the intention distribution $p(i|s,a)$.

## 5  Implementation

In this section, we discuss implementation details that can alleviate instability of the training procedure of our model. The first indicator that the training has become unstable is a high classification accuracy of the discriminator. In this case, it is difficult for the generator to produce a meaningful policy as the reward signal from the discriminator is flat and the TRPO gradient of the generator vanishes. In an extreme case, the discriminator assigns all the generator samples to the same class and it is impossible for TRPO to provide a useful gradient as all generator samples receive the same reward. Previous work suggests several ways to avoid this behavior. These include leveraging the Wasserstein distance metric to improve the convergence behavior [3] and adding *instance noise* to the inputs of the discriminator to avoid degenerate generative distributions [32]. We find that adding the Gaussian noise helped us the most to control the performance of the discriminator and to produce a smooth reward signal for the generator policy. During our experiments, we anneal the noise similar to [32], as the generator policy improves towards the end of the training.

An important indicator that the generator policy distribution has collapsed to a uni-modal policy is a high or increasing loss of the intention-prediction network $p(i|s,a)$. This means that the prediction of the latent variable $i$ is difficult and consequently, the policy behavior can not be categorized into separate skills. Hence, the policy executes the same skill for different values of the latent variable. To prevent this, one can increase the weight of the latent intention cost $\lambda_I$ in the generator loss or add more instance noise to the discriminator, which makes its reward signal relatively weaker.

In this work, we employ both categorical and continuous latent variables to represent the latent intention. The advantage of using a continuous variable is that we do not have to specify the number of possible values in advance as with the categorical variable and it leaves more room for interpolation between different skills. We use a softmax layer to represent categorical latent variables, and use a uniform distribution for continuous latent variables as proposed in [7].

## 6  Experiments

Our experiments aim to answer the following questions: (1) Can we segment unstructured and unlabelled demonstrations into skills and learn a multi-modal policy that imitates them? (2) What

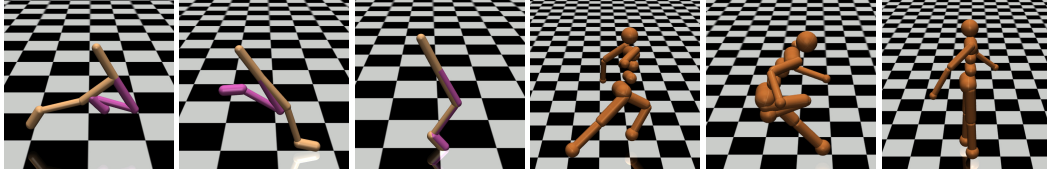

Figure 1: *Left:* Walker-2D running forwards, running backwards, jumping. *Right:* Humanoid running forwards, running backwards, balancing.

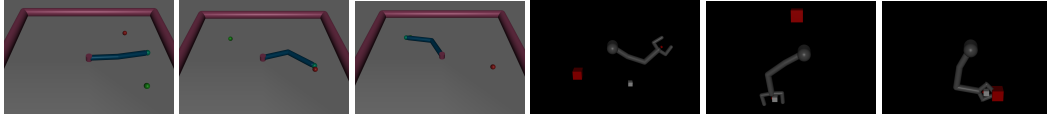

Figure 2: *Left:* Reacher with 2 targets: random initial state, reaching one target, reaching another target. *Right:* Gripper-pusher: random initial state, grasping policy, pushing (when grasped) policy.

is the influence of the introduced intention-prediction cost on the resulting policies? (3) Can we autonomously discover the number of skills presented in the demonstrations, and even accomplish them in different ways? (4) Does the presented method scale to high-dimensional policies? (5) Can we use the proposed method for learning hierarchical policies? We evaluate our method on a series of challenging simulated robotics tasks described below. We would like to emphasize that the demonstrations consist of shuffled state-action pairs such that no temporal information or segmentation is used during learning. The performance of our method can be seen in our supplementary video[2].

## 6.1 Task setup

**Reacher** The Reacher environment is depicted in Fig. 2 (left). The actuator is a 2-DoF arm attached at the center of the scene. There are several targets placed at random positions throughout the environment. The goal of the task is, given a data set of reaching motions to random targets, to discover the dependency of the target selection on the intention and learn a policy that is capable of reaching different targets based on the specified intention input. We evaluate the performance of our framework on environments with 1, 2 and 4 targets.

**Walker-2D** The Walker-2D (Fig. 1 left) is a 6-DoF bipedal robot consisting of two legs and feet attached to a common base. The goal of this task is to learn a policy that can switch between three different behaviors dependent on the discovered intentions: running forward, running backward and jumping. We use TRPO to train single expert policies and create a combined data set of all three behaviors that is used to train a multi-modal policy using our imitation framework.

**Humanoid** Humanoid (Fig. 1 right) is a high-dimensional robot with 17 degrees of freedom. Similar to Walker-2D the goal of the task is to be able to discover three different policies: running forward, running backward and balancing, from the combined expert demonstrations of all of them.

**Gripper-pusher** This task involves controlling a 4-DoF arm with an actuated gripper to push a sliding block to a specified goal area (Fig. 2 right). We provide separate expert demonstrations of grasping the object, and pushing it towards the goal starting from the object already being inside the hand. The initial positions of the arm, block and the goal area are randomly sampled at the beginning of each episode. The goal of our framework is to discover both intentions and the hierarchical structure of the task from a combined set of demonstrations.

## 6.2 Multi-Target Imitation Learning

Our goal here is to analyze the ability of our method to segment and imitate policies that perform the same task for different targets. To this end, we first evaluate the influence of the latent intention cost on the Reacher task with 2 and 4 targets. For both experiments, we use either a categorical intention distribution with the number of categories equal to the number of targets or a continuous,

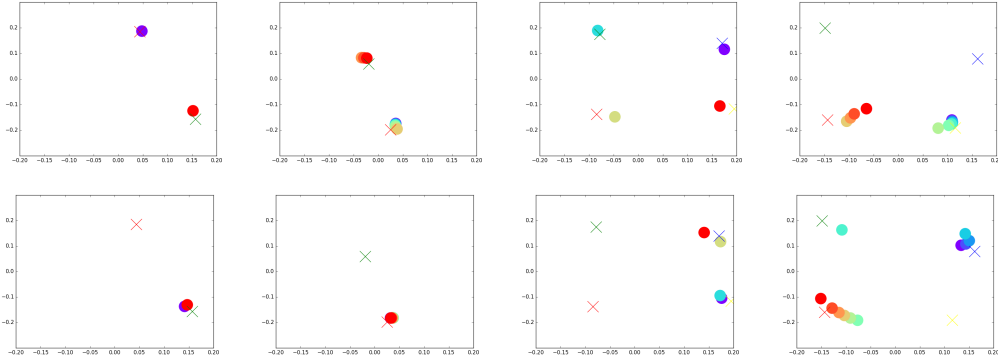

Figure 3: Results of the imitation GAN with (top row) and without (bottom row) the latent intention cost. *Left:* Reacher with 2 targets(crosses): final positions of the reacher (circles) for categorical (1) and continuous (2) latent intention variable. *Right:* Reacher with 4 targets(crosses): final positions of the reacher (circles) for categorical (3) and continuous (4) latent intention variable.

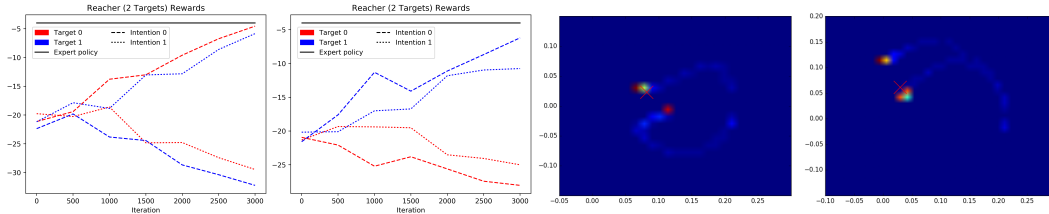

Figure 4: *Left:* Rewards of different Reacher policies for 2 targets for different intention values over the training iterations with (1) and without (2) the latent intention cost. *Right:* Two examples of a heatmap for 1 target Reacher using two latent intentions each.

uniformly-distributed intention variable, which means that the network has to discover the number of intentions autonomously. Fig. 3 top shows the results of the reaching tasks using the latent intention cost for 2 and 4 targets with different latent intention distributions. For the continuous latent variable, we show a span of different intentions between -1 and 1 in the 0.2 intervals. The colors indicate the intention "value". In the categorical distribution case, we are able to learn a multi-modal policy that can reach all the targets dependent on the given latent intention (Fig. 3-1 and Fig. 3-3 top). The continuous latent intention is able to discover two modes in case of two targets (Fig. 3-2 top) but it collapses to only two modes in the four targets case (Fig. 3-4 top) as this is a significantly more difficult task.

As a baseline, we present the results of the Reacher task achieved by the standard GAN imitation learning presented in [16] without the latent intention cost. The obtained results are presented in Fig. 3 bottom. Since the network is not encouraged to discover different skills through the intention learning cost, it collapses to a single target for 2 targets in both the continuous and discrete latent intention variables. In the case of 4 targets, the network collapses to 2 modes, which can be explained by the fact that even without the latent intention cost the imitation network tries to imitate most of the presented demonstrations. Since the demonstration set is very diverse in this case, the network learned two modes without the explicit instruction (latent intention cost) to do so.

To demonstrate the development of different intentions, in Fig. 4 (left) we present the Reacher rewards over training iterations for different intention variables. When the latent intention cost is included, (Fig. 4-1), the separation of different skills for different intentions starts to emerge around the 1000-th iteration and leads to a multi-modal policy that, given the intention value, consistently reaches the target associated with that intention. In the case of the standard imitation learning GAN setup (Fig. 4-2), the network learns how to imitate reaching only one of the targets for both intention values.

In order to analyze the ability to discover different ways to accomplish the same task, we use our framework with the categorical latent intention in the Reacher environment with a single target.

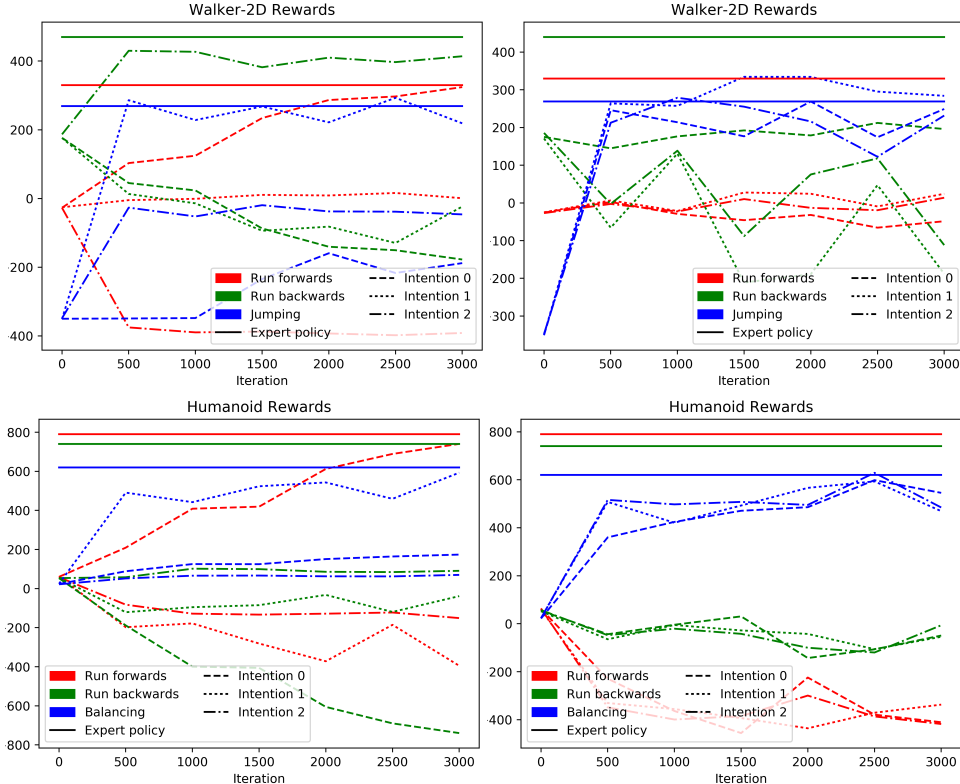

Figure 5: *Top:* Rewards of Walker-2D policies for different intention values over the training iterations with (left) and without (right) the latent intention cost. *Bottom:* Rewards of Humanoid policies for different intention values over the training iterations with (left) and without (right) the latent intention cost.

Since we only have a single set of expert trajectories that reach the goal in one, consistent manner, we subsample the expert state-action pairs to ease the intention learning process for the generator. Fig. 4 (right) shows two examples of a heatmap of the visited end-effector states accumulated for two different values of the intention variable. For both cases, the task is executed correctly, the robot reaches the target, but it achieves it using different trajectories. These trajectories naturally emerged through the latent intention cost as it encourages different behaviors for different latent intentions. It is worth noting that the presented behavior can be also replicated for multiple targets if the number of categories in the categorical distribution of the latent intention exceeds the number of targets.

### 6.3 Multi-Task Imitation Learning

We also seek to further understand whether our model extends to segmenting and imitating policies that perform different tasks. In particular, we evaluate whether our framework is able to learn a multi-modal policy on the Walker-2D task. We mix three different policies – running backwards, running forwards, and jumping – into one expert policy $\pi_E$ and try to recover all of them through our method. The results are depicted in Fig. 5 (top). The additional latent intention cost results in a policy that is able to autonomously segment and mimic all three behaviors and achieve a similar performance to the expert policies (Fig. 5 top-left). Different intention variable values correspond to different expert policies: 0 - running forwards, 1 - jumping, and 2 - running backwards. The imitation learning GAN method is shown as a baseline in Fig. 5 (top-right). The results show that the policy collapses to a single mode, where all different intention variable values correspond to the jumping behavior, ignoring the demonstrations of the other two skills.

To test if our multi-modal imitation learning framework scales to high-dimensional tasks, we evaluate it in the Humanoid environment. The expert policy is constructed using three expert policies: running backwards, running forwards, and balancing while standing upright. Fig. 5 (bottom) shows the rewards obtained for different values of the intention variable. Similarly to Walker-2D, the latent

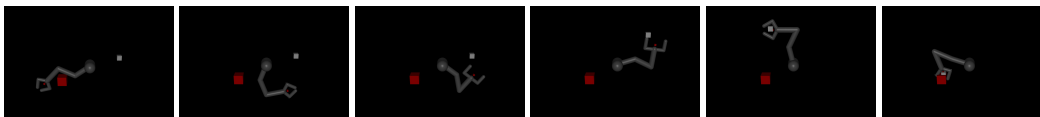

Figure 6: Time-lapse of the learned Gripper-pusher policy. The intention variable is changed manually in the fifth screenshot, once the grasping policy has grasped the block.

intention cost enables the neural network to segment the tasks and learn a multi-modal imitation policy. In this case, however, due to the high dimensionality of the task, the resulting policy is able to mimic running forwards and balancing policies almost as well as the experts, but it achieves a suboptimal performance on the running backwards task (Fig. 5 bottom-left). The imitation learning GAN baseline collapses to a uni-modal policy that maps all the intention values to a balancing behavior (Fig. 5 bottom-right).

Finally, we evaluate the ability of our method to discover options in hierarchical IRL tasks. In order to test this, we collect expert policies in the Gripper-pusher environment that consist of grasping and pushing when the object is grasped demonstrations. The goal of this task is to check whether our method will be able to segment the mix of expert policies into separate grasping and pushing-when-grasped skills. Since the two sub-tasks start from different initial conditions, we cannot present the results in the same form as for the previous tasks. Instead, we present a time-lapse of the learned multi-modal policy (see Fig. 6) that presents the ability to change in the intention during the execution. The categorical intention variable is manually changed after the block is grasped. The intention change results in switching to a pushing policy that brings the block into the goal region. We present this setup as an example of extracting different options from the expert policies that can be further used in an hierarchical reinforcement learning task to learn the best switching strategy.

# 7    Conclusions

We present a novel imitation learning method that learns a multi-modal stochastic policy, which is able to imitate a number of automatically segmented tasks using a set of unstructured and unlabeled demonstrations. The presented approach learns the notion of intention and is able to perform different tasks based on the policy intention input. We evaluated our method on a set of simulation scenarios where we show that it is able to segment the demonstrations into different tasks and to learn a multi-modal policy that imitates all of the segmented skills. We also compared our method to a baseline approach that performs imitation learning without explicitly separating the tasks.

In the future work, we plan to focus on autonomous discovery of the number of tasks in the given pool of demonstrations as well as evaluating this method on real robots. We also plan to learn an additional hierarchical policy over the discovered intentions as an extension of this work.

## Acknowledgements

This research was supported in part by National Science Foundation grants IIS-1205249, IIS-1017134, EECS-0926052, the Office of Naval Research, the Okawa Foundation, and the Max-Planck-Society. Any opinions, findings, and conclusions or recommendations expressed in this material are those of the author(s) and do not necessarily reflect the views of the funding organizations.

## Footnotes

[2]http://sites.google.com/view/nips17intentiongan

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
