[Reviews · NeurIPS 2017]

Reviewer 1



The paper describes a new learning model able to discover 'intentions' from expert policies by using an imitation learning framework. The idea is mainly based on the GAIL model which aims at learning by imitation a policy using a GAN approach. The main difference in the article is that the learned policy is, in fact, a mixture of sub-policies, each sub-policy aiming at automatically matching a particular intention in the expert behavior. The GAIL algorithm is thus derived with this mixture, resulting in an effective learning technique. Another approach is also proposed where the intention will be captured through a latent vector by derivating the InfoGAN algorithm for this particular case. Experiments are made on 4 different settings and show that the model is able to discover the underlying intentions contained in the demonstration trajectories. Comments: First of all, the task focused in the paper is interesting. Discovering 'intentions' and learning sub-policies is a key RL problem that has been the source of many old and new papers. The proposed models are simple but effective extensions of existing models. Actually, I do not really understand what is the difference made by the authors between 'intentions' and 'options' in the literature. It seems to me that intentions are more restricted than options since there are no natural switching mechanisms in the proposed approach while options models are able to choose which option to use at any time. Moreover, w.r.t options literature, the main originality of the paper is both in how the intentions are discovered (using GAIL), and also in the fact that this is made by imitation while many options learning models are based on a classical RL setting. A discussion on this point is important, both in the 'related work' part, but also in the experimental part (I will go back to this point later in the review). At the end, it seems that the model discovered sub-policies, but is still unable to know how to assemble these sub-policies in order to solve complex problems. So the proposed model is more a first step to learn an efficient agent than a complete solution. (This is for example illustrated in Section 6 "[...]the categorical intention variable is manually changed [...]"). The question is thus what can be done with the learned policy ? How the result of the algorithm will be used ? Concerning the presentation of the model, the notations are not clear. Mainly, \pi^i gives the impression that the policy is indexed by the intention i (which is not the case, since, as far as I understand, the indexation by the intention is in fact contained in the notation $\pi(a|s,i)$) . Moreover, the section concerning the definition of the reward is unclear: as far as I understand, in your setting, trajectories are augmented with the intention value at each timestep. But this intention value i_t is not used during learning, but will be used for evaluating the reward value. I think that this has to be made more clear in the paper if accepted. Concerning the experimental sections: * First, the environments are quite simple, and only focused on fully-observable MDP. The extension of the proposed model to PO-MDP could be discussed in the paper (since I suppose it is not trivial). * Second, there is no comparison of the proposed approach with options discovery/hierarchical models techniques like "Imitation Learning with Hierarchical Actions -- Abram L. Friesen and Rajesh P. N. Rao", or even "Active Imitation Learning of Hierarchical Policies Mandana Hamidi, Prasad Tadepalli, Robby Goetschalckx, Alan Fern". This is a clear weak point of the paper. * I do not really understand how the quantitative evaluation is made (reward). In order to evaluate the quality of each sub-policy w.r.t each intention, I suppose that a manual matching is made between the value of $i$ and the 'real' intention. Could you please explain better explain that point ? * The difference between the categorical intentions and continuous one is not well discussed and evaluated. Particularly, on the continuous case, the paper would gain if the interpolation between sub-policies is evaluated as it is suggested in the article. Conclusion: An interesting extension of existing models, but with some unclear aspects, and with an experimental section that could be strenghtened

Reviewer 2



This paper proposes to learn stills from demonstrations without any a priori knowledge. Using GAN, it generates trajectories from a mixture of policies, and imitate the demonstration. This idea is very simple and easy to be understood. I accept that the idea could works. There are however some issues in the details. The objective Eq.7 does not show optimizing p(i), which is surely to be optimized. In Eq.7, should not the laster term E log(p(i)) be expressed as H(p(i)), instead of H(i)? And the former is not a constant. Figure 5 is too small, and too crowd to read clearly.

Reviewer 3



- Summary This paper considers a multi-task imitation learning problem where the agent should learn to imitate multiple expert policies without having access to the identity of the tasks. The proposed method is based on GAIL (generative adversarial imitation learning) with an additional objective that encourages some of the latent variables (called intention variables) to be easily inferred from the generated trajectories. The results show that the intention variables can captures different modes of the expert behaviors on several Mujoco tasks. [Pros] - The problem considered in this paper is interesting. - The proposed objective is novel, though it is very similar to InfoGAN or GAIL + [Florensa et al.]'s idea in the end. [Cons] - (minor) The experimental results are good but not "impressive" because the expert behaviors are quite clearly separated (see Quality for details). - (minor) The arguments on hierarchical RL are not much convincing (see Quality for details). - Quality The proposed method and its connection to InfoGAN are interesting and sound reasonable. Some suggestions to improve the paper: 1) The expert behaviors seem quite well-separated in the experiment (e.g., walking forward or backward). It could be much more impressive if the paper showed that the proposed model can capture subtle but distinguishable differences of behaviors (e.g., different styles of walking forward) in separate latent codes. 2) "Hierarchical RL" part is not much convincing. The goal of option discovery in HRL is to find useful temporal abstractions from "temporally unsegmented" tasks. A similar problem formulation in imitation learning context would be to show two subtasks (grasping -> pushing) in a "single" episode and let the agent to figure out how to temporally segment the expert's behavior into two subtasks and learn them. In the experiment, however, the authors gave two subtasks as separate episodes to my understanding, and the agent only needed to correctly model two behaviors into two intention variables, which is a much easier problem than the original problem of "option discovery". 3) It would be interesting to show the model's behaviors from interpolated or extrapolated intention variables. 4) The proposed method is actually quite close to [Florensa et al.]'s method in the sense that the latent variables are encouraged to be inferred from the agent's behavior, though they did not focus on imitation learning. It would be good to make a connection to this work in more detail. - Clarity The paper is well-written except for the following: 1) In Line 25-32, the motivating example can be misleading. The paper claims that it aims to learn skills from unstructured data (e.g., learning to grasp, reach, cut from a video showing cooking). It sounds like this paper aims to discover subtasks from temporally unsegmented data, which is not the case because the paper provided temporally segmented (but unlabeled) data to the model. This is also related to the comment on HRL in the Quality section. 2) In Line 33-35, the term "automatically segmented tasks" or "unstructured demonstrations" are again a bit misleading as discussed above. Instead, "unlabeled demonstrations" is more clear. - Originality The multi-task imitation learning problem is interesting, and the proposed idea of using GAIL with InfoGAN-like objective is novel. - Significance Multi-task imitation learning is an important research direction. This paper shows a nice application of GAIL and InfoGAN-like approach to capture the underlying modes of diverse behaviors into disentangled latent variables. Though some of the statements are a bit misleading, I think this is overall a solid paper.